# Consideration of the Heating of High-Performance Concretes during Cyclic Tests in the Evaluation of Results

**Melchior Deutscher** 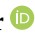

Institute of Concrete Structures, Technische Universität Dresden, 01069 Dresden, Germany;
Melchior.Deutscher@tu-dresden.de; Tel.: +49-351-463-40473

**Abstract:** Material-efficient, highly load-bearing members made of high-performance compressive concretes are often exposed to cyclical loads because of their slender construction, which can be relevant to the design. When investigating the fatigue behaviour of high-performance concretes in pressure swell tests, however, the specimen temperature rises strongly owing to the elevated loading rate at frequencies higher than 3 Hz. This leads to a negative influence on the achieved number of load cycles compared to tests carried out at slow speeds and calculated values, for example, according to fib Model Code 2010. This phenomenon, which was already observed, must be considered when generating design formulae or Wöhler lines for component design, as the test conditions with high constant load frequencies as well as sample storage in a climate chamber at constant conditions are prerequisites that cannot be expected in real material applications. Therefore, laboratory testing influences must be eliminated in order to avoid underestimating the material. Instead of adjusting the test conditions to prevent or control temperature development, as was the case in previous approaches, this article shows how the temperature effects can be corrected when analysing the results, considering both the applied stress and the maximum temperature reached. For this purpose, a calculation method was developed that was validated on the basis of a large number of fatigue tests. Thus, in the future, the application of one temperature sensor to the test specimen can effectively advance the extraction of values for Wöhler curves, even with high test frequencies.

**Keywords:** UHPC; fatigue behaviour; temperature increase; Wöhler curve

## 1. Introduction

By developing an optimised packing density and achieving a low water–cement ratio through the use of very fine-grained admixtures like silica fume and high-performance superplasticisers, it is possible to achieve concretes with compressive strengths of up to 200 MPa, see, e.g., [1,2].

In general, greater slimness can be achieved, for example, in long-span bridges or wind turbines with increasing material strength. Such constructions are exposed to very high numbers of load cycles and are susceptible to vibration because of the increasing slenderness. A comprehensive knowledge of the fatigue standard is therefore particularly important for high and ultra-high-performance concretes. Usually, standardising pressure swell tests with high loading frequencies are carried out to investigate fatigue strength. In several research papers, a temperature increase in the sample was measured during testing [3–9], which usually resulted in an earlier failure, compared to the expected values according to fib Model Code 2010 [10] or to slowly cycled attempts [7,9,11–15]. In the recent past, detailed parameter studies have been carried out in various parallel research projects [11–18]. For one UHPC, the load-related parameters such as the relation between lower and upper stress levels and frequency as well as the parameters of maximum grain size, concrete strength, and concrete age were investigated regarding concrete heating, as in the project presented here and already published in [13,16,19–21].

At temperatures of up to 90 °C observed so far under fatigue loading [13,21], heating led to a reduction in the static compressive strength. In [22] this influence on the static

compressive strength was shown in addition to the fib Model Code 2010 [10] for the analysed UHPC. It was also evident that the observed effect was particularly strong for specimens stored in a typical laboratory climate chamber (20 °C and 65% RH) until testing. Consequently, this reduction was found to be a test-related influence that cannot be expected in the real building construction.

The force-controlled load values for the maximum and minimum stress during a fatigue test remain constant over the duration of the test. If the concrete strength decreases from an increase in temperature under cyclic loading, the related load range increases at the same time. Thus, this effect leads to an underestimation of the fatigue strength. Heating during cyclic tests can be prevented as far as possible simply by greatly reducing the test frequency. However, this would disproportionately increase the duration for cyclic tests for standardising the fatigue strength of UHPC. A different approach for effective material testing is therefore required and will be presented in this paper. The aim of the presented method is evaluating the test results as a function of the temperature development, so that high frequencies have no negative influence on the results. In contrast to [15], the test procedure does not have to be adapted in order to eliminate the temperature effect.

## 2. Database and Methods

### 2.1. Specimen Material, Geometry, Application and Test Setup

The basis for the evaluation method presented below is a database, most of which has already been published in [13,16,19–21]. The investigations were carried out on a UHPC, which is the reference material in the German Priority Programme SPP 2020 [23] and was therefore also used in other research works [24,25]. The current European standardisation (DIN EN 1992-1-1) [26] classifies concrete up to a strength class of C90/105. The prenormative fib Model Code 2010 [10] additionally includes the strength class C100/115. From strength class C55/67, the term high-performance concrete (HPC) is used [26]. The designation ultra-high-performance concrete (UHPC) is used for even (significantly) higher strengths, but there is no generally accepted international definition. In the presented research, the term UHPC is used for concretes with compressive strengths above class C100/115. Detailed information on the material can be found, e.g., in [13]. Table 1 shows only the essential material properties, including the strengths of the individual concrete batches for the cycling tests.

**Table 1.** Material properties of UHPC.

| Property | Unit | Value |
|---|---|---|
| Mean compressive strength after 28 days | MPa | 162.7 |
| Mean compressive strength after 90 days | MPa | 182.7 |
| Modulus of elasticity | MPa | 47,050 |
| Reference strength batch a | MPa | 182.7 |
| Reference strength batch b | MPa | 182.4 |
| Reference strength batch c1 | MPa | 188.5 |
| Reference strength batch c2 | MPa | 193.3 |

The reference strength values per batch were each determined on three test specimens that had the same geometry as the samples for the cyclic tests and a comparable age. Batch c was tested partly in Dresden ($c_1$) and partly in Weimar ($c_2$). It should be noted that there were more than 6 months between these tests.

The test procedure and specimen geometry were based on [27] for all tests. Cylinders with a diameter of 60 mm and a height of 180 mm were used. Details about the storage conditions, the specimen's preparation, and the application of measuring equipment can be found in [13,20]. All tested samples had a minimum concrete age of 90 days. In [28] it was shown that at that point the strength development was almost completed and no influence on the related stresses were to be expected.

Information about the testing machine used in the lab in Dresden can also be found in [13]. Only the tests at higher than 20 Hz were carried out in Weimar (Germany) at the testing laboratory of the Institute of Structural Engineering. There, a servo-hydraulic testing machine (Figure 1) with frequency of up to 50 Hz at a maximum cyclic load of 800 MPa was used.

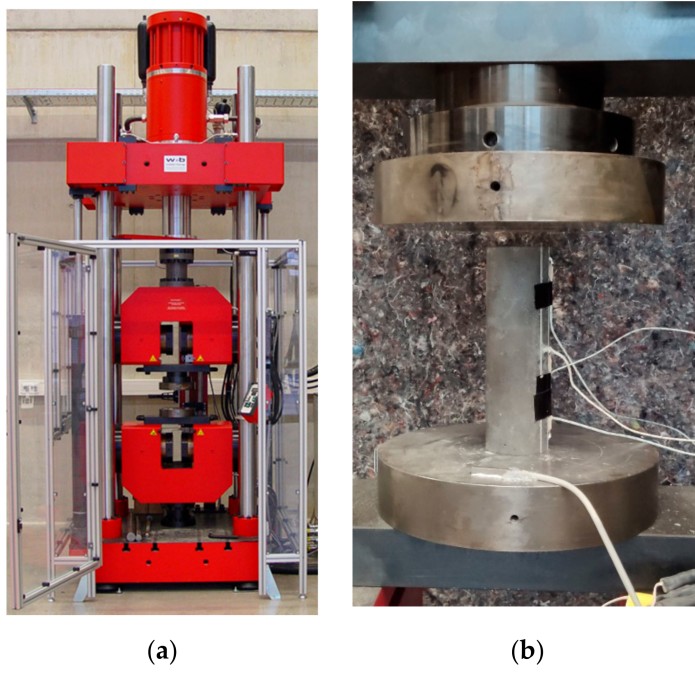

(**a**) (**b**)

**Figure 1.** (**a**) Servo-hydraulic testing machine in the laboratory in Weimar; (**b**) UHPC sample in this testing machine.

### 2.2. Database

The previous tests are summarised in Table 2. Three fatigue tests were carried out for each load configuration in the course of the project. In addition to the key input parameters frequency $f$ and load range $S_{c,i}$, the main results—maximum temperature $T_{max}$ and number of load cycles achieved ($N$)—are also shown, which are important for the evaluation. The maximum temperature values were measured on the concrete surface at half the specimen height. All stress levels are given as related input values in form of the quotient between the upper $S_{c,max}$ and lower $S_{c,min}$ load of a load cycle to the average concrete compressive strength of the batch $f_{cm}$. The upper limit for the load change had previously been set at 2 million. "Run-throughs" were therefore stopped at 2 million load changes and not considered in the evaluation in relation to the number of load cycles. In all these cases, the temperature development was completed.

**Table 2.** Test results that were available as a basis and for the verification of the evaluation method.

| Designation | Frequency $f$ (Hz) | $S_{c,max} = f_{c,max}/f_{cm}$ (—) | $S_{c,min} = f_{c,min}/f_{cm}$ (—) | $\Delta S$ (—) | $T_{max}$ (°C) | Number of Load Cycles $N$ (—)[1] | Published in | Batch |
|---|---|---|---|---|---|---|---|---|
| S1T S4 S5 | 3 | 0.60 | 0.10 | 0.50 | 26.9 27.1 25.0 | 2,000,000 2,000,000 2,000,000 | [13,21] | a |
| S3T S8 S9 | 3 | 0.70 | 0.10 | 0.60 | 30.4 31.5 32.9 | 198,142 2,000,000 2,000,000 | [13,21] | a |
| S5T S12 S13 | 3 | 0.80 | 0.10 | 0.70 | 31.6 42.3 36.2 | 8956 28,419 9954 | [13,21] | a |
| S5T S18 S19 | 7.5 | 0.70 | 0.10 | 0.60 | 48.4 56.3 63.2 | 56,600 61,687 178,608 | [21] | b |
| S2T S6 S7 | 10 | 0.60 | 0.10 | 0.50 | 34.9 39.9 33.0 | 2,000,000 2,000,000 2,000,000 | [13,21] | a |
| S1T S18 S19 | 10 | 0.65 | 0.05 | 0.60 | 79.0 74.4 63.8 | 77,790 71,994 51,360 | - | $c_1$ |
| S13 S14 S15 | 10 | 0.65 | 0.10 | 0.55 | 50.4 52.7 55.3 | 71,193 337,423 186,558 | [21] | b |
| S5T S12 S13 | 10 | 0.65 | 0.15 | 0.50 | 44.0 40.0 39.9 | 2,000,000 2,000,000 2,000,000 | - | $c_1$ |
| S2T S6 S7 | 10 | 0.70 | 0.05 | 0.65 | 63.6 58.3 77.9 | 32,902 28,041 44,653 | - | $c_1$ |
| S11 S4T S14T | 10 | 0.70 | 0.10 | 0.60 | 65.9 66.4 60.3 | 50,583 54,385 64,970 | - | $c_1$ $c_2$ |
| S4T S10 S11 | | | | | 64.0 58.1 53.0 | 122,095 77,592 49,057 | [13,21] | a |
| S6T S14 S15 | 10 | 0.70 | 0.15 | 0.55 | 54.1 53.5 67.9 | 135,598 80,846 157,294 | - | $c_1$ |
| S3T S8 S9 | 10 | 0.75 | 0.05 | 0.70 | 52.2 62.3 37.3 | 18,160 15,255 6830 | - | $c_1$ |
| S4T S26 S27 | 10 | 0.75 | 0.10 | 0.65 | 63.5 60.6 56.5 | 40,903 37,487 28,062 | [21] | b |
| S7T S16 S17 | 10 | 0.75 | 0.15 | 0.60 | 27.1 51.9 54.1 | 7491 46,398 46,502 | - | $c_1$ |
| S6T S14 S15 | 10 | 0.80 | 0.10 | 0.70 | 51.2 23.5 44.1 | 23,413 414[2] 15,904 | [13,21] | a |
| S6T S20 S21 | 12.5 | 0.70 | 0.10 | 0.60 | 85.3 77.1 80.2 | 141,471 86,184 119,206 | [21] | b |
| S7T S16 S17 | 20 | 0.60 | 0.10 | 0.50 | 53.5 57.1 55.9 | 2,000,000 180,444 2,000,000 | [13,21] | a |
| S8T S18 S19 | 20 | 0.70 | 0.10 | 0.60 | 63.8 91.8 49.2 | 70,702 86,759 45,535 | [13,21] | a |
| S9T S20 S21 | 20 | 0.80 | 0.10 | 0.70 | 57.0 57.1 28.8 | 26,213 24,233 5890 | [13,21] | a |
| S11T S31 | 25 | 0.70 | 0.10 | 0.60 | 57.5 75.4 | 39,266 49,987 | - | $c_2$ |
| S13T S24 S25 | 30 | 0.70 | 0.10 | 0.60 | 75.0 45.7 39.7 | 58,735 29,300 22,310 | - | $c_2$ |

[1] test with *2 million* load cycles were interrupted (no failure under cyclic loading), [2] large void in the load introduction area.

### 2.3. Number of Load Changes without Temperature Consideration

Figure 2 shows all test results in a maximum stress-level number of cycles diagram (compare [13,21]). The number of load cycles is plotted logarithmically. Based on many experiments, the generation of a trend line allowed us to create a Wöhler curve. The fib Model Code 2010 [10], in which a mathematical description for the Wöhler curve is given, was used for the calculated expectation values also added to the diagram.

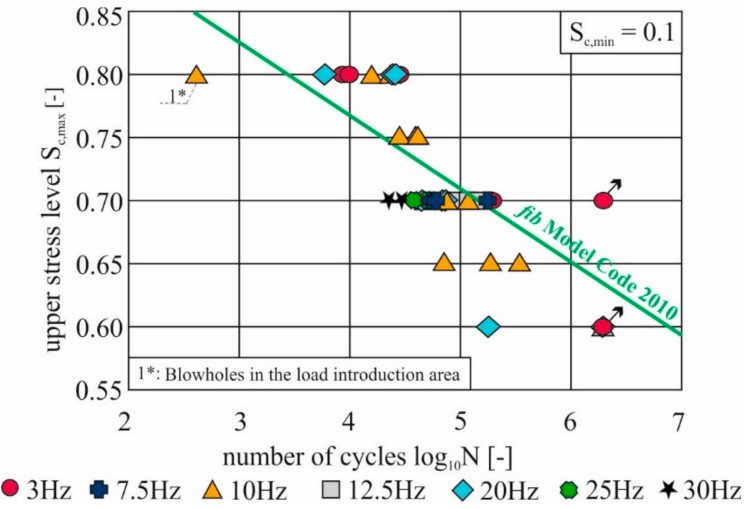

**Figure 2.** Achieved numbers of load cycles at different maximum stress levels and load frequencies.

Especially in the range of 0.65 to 0.70 as the maximum stress level, a significant drop below the values according to fib Model Code 2010 [10] was observed. In [13,21], a clear temperature increase until failure was observed for these tests. Only the attempts at 3 Hz with almost no heating showed no undercut. Solely at a stress level below 0.55, run-throughs were almost exclusively generated, independent of frequency.

### 2.4. Temperature Effect on Static Compressive Strength

In [22], a substantial study on the effect of temperature $T$ on the compressive strength of the investigated UHPC was presented. Specimen geometry, sample age, and test setup were basically the same. The compressive strengths $f_c(T)$ were determined in a temperature range between $-10$ and $90\,^{\circ}$C. Before the tests, the samples were cooled down in a deep freezer or heated up in an oven. The results at room temperature up to $90\,^{\circ}$C covered the relevant temperature range to which the material was exposed during cyclic pressure swell tests. Figure 3 shows the experimental results from 20 to $90\,^{\circ}$C relevant to the developed evaluation method and a linear approximation that corresponded to the approximation according to fib Model Code 2010 [10]. The reference value was the average concrete strength $f_{cm}$ under standard conditions. With an increase in the material temperature, the compressive strength $f_c(T)$ decreased.

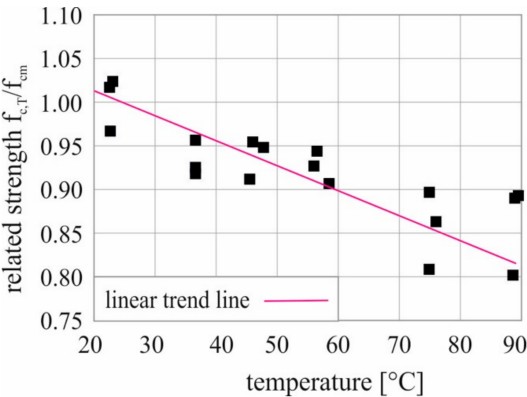

**Figure 3.** Related static compressive strength of UHPC at material temperatures between 20 and 90 °C with linear trend line [22].

The linear trend line describing the temperature-dependent compressive strength $f_c(T)$ is shown in Equation (1).

$$f_c(T)/f_{c,m} = -0.0029 \cdot T + 1.0718. \tag{1}$$

### 2.5. Consideration of Temperature during the Cyclic Tests

At the beginning of the cyclic tests, the samples were at ambient temperature for which the relative material strength was 1.00 (see Figure 3). During the cyclic tests with frequencies from 7.5 Hz onwards, the concrete temperature increased so that the relative material strength decreased. Using the equation from Figure 3, the relative strength could now be plotted as a function of the temperature variation over the duration of the test. Figure 4 shows the change in the uniaxial concrete strength with regard to the observed material heating during the cyclic tests. In the diagram, one example is shown for each group of tests with a constant minimum stress level of 0.10 and a varying maximum stress level from 0.60 to 0.80. The frequency was 10 Hz in each of the tests shown.

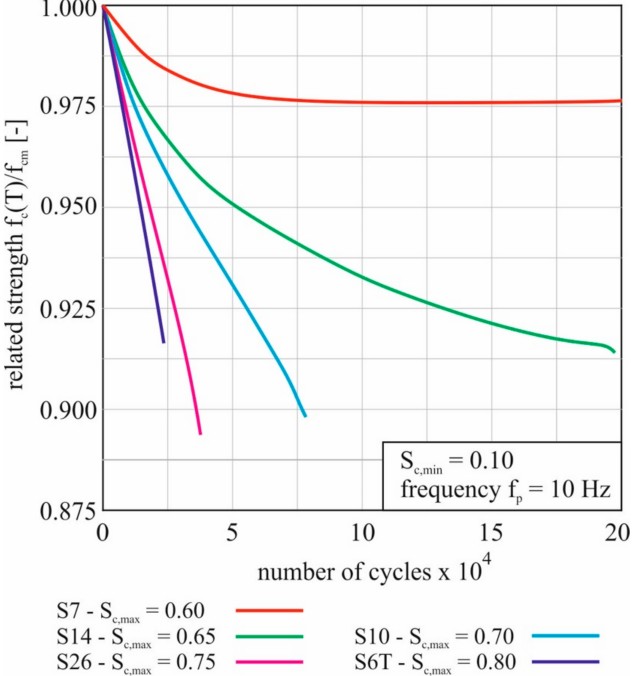

**Figure 4.** Decreasing relative concrete compressive strengths in fatigue tests dependent on the occurring heating of the samples.

The diagram shows the number of load cycles up to 200,000. The trial S7 ran up to 2 million load changes. Therefore, in the diagram only the first 10% of this course can be seen. However, the temperature and thus the strength did not change significantly until the test was stopped.

The curves are characteristic of the measured temperature curves in the same samples shown in [13,16,19,28], only in negative instead of positive directions. It can be seen that the material strength was reduced by less than 2.5% in the run-through attempt S7. In the tests with stronger heating up to failure, a reduction of around 10% was achieved. If the material strength decreased, the "real" minimum and maximum stresses increased over time with cyclic force application because the "real" reference strength changed, but the nominal stress limits applied in the tests remained constant. This is described by Equation (2):

$$S_{c,max/min}(T) = S_{c,max/min}/(f_c(T)/f_{c,m}). \tag{2}$$

The temperature-dependent change in the upper and lower stress levels $S_{c,min/max}(T)$ is illustrated with dashed lines in Figure 5 for the tests presented in Figure 4. Additionally, the stress play $\Delta S_c(T)$ according to Equation (3) is shown considering the changed reference strengths of the material (solid lines).

$$\Delta S_c(T) = S_{c,max} - S_{c,min}. \tag{3}$$

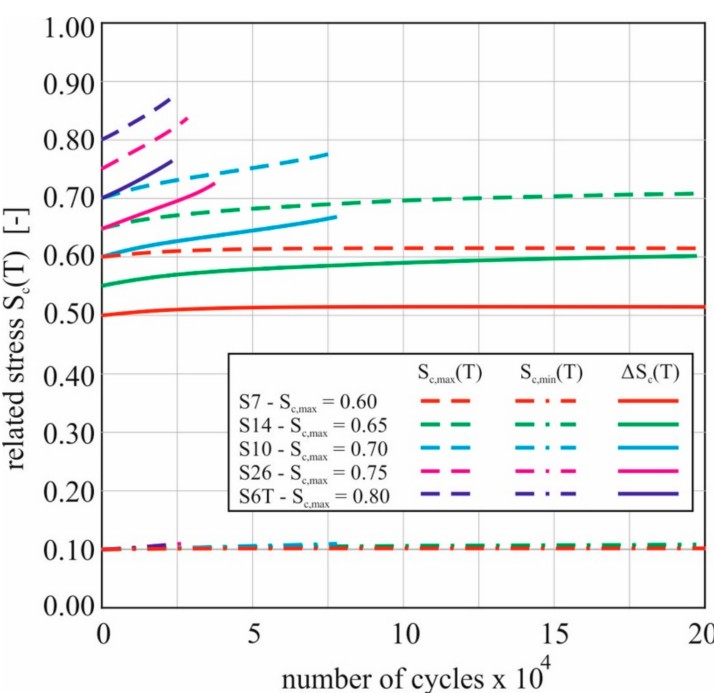

**Figure 5.** Development of the maximum and minimum stress levels and the stress range during the cyclic tests as a function of temperature and, thus, the changing reference strength of concrete.

Basically, all graphs increased over the duration of the test. Since the maximum and minimum stresses changed by the same percentage depending on the related strength, the absolute change was also smaller with a lower input value. This can be seen especially in the change in the maximum stress compared to the minimum stress. For example, the upper stress increased by more than 8% in test S26, while the lower voltage only increased by 1%. This resulted in a stress range for the test specimen that increased almost parallel to the maximum stress level. As the temperature rose, the energy input increased and so did the damage per load cycle.

### 2.6. Development of the Evaluation Method

The presented results allowed the assumption that the temperature was a reason for the premature failure of specimens in cyclic tests with high frequency because it was shown that with increasing material temperature, the relative cyclic fatigue stress also increased. This in turn led to a reduction in the expected number of load cycles according to fib Model Code 2010 [10]. However, a classification of cyclic tests in a Wöhler curve diagram is only possible for exactly one stress range. Influences on the tested material that go beyond pure repeated mechanical stress cannot be considered. It is therefore necessary to define a corrected stress level based on the initial stress and the measured temperature, with which the temperature-dependent degradation of the compressive strength can be taken into account sufficiently precisely. It is important to ensure that the reduction is not too large in order to avoid overestimating the fatigue strength of the concrete. It should also be a user-friendly solution that does not significantly complicate the test procedure in the lab and the data evaluation.

Because of the very small change in the minimum stress level $S_{c,min}$, this influence was neglected in the evaluation, since classification in a Wöhler curve diagram always requires either a constant maximum or a constant minimum stress level. In a first step, the maximum temperature per test $T_{max}$ (see Table 1) and the average temperature calculated over the test duration were used as constant values for correction. In both cases, the relative strength $f_c(T)/f_{cm}$ was determined with Equation (1), and the related maximum stress level $S_{c,max}$ with Equation (2). The results can be seen in Figure 6 for all tests with a frequency of 10 Hz and a minimum stress level $S_{c,min} = 0.1$. The attempts with a maximum stress level $S_{c,max}$ of 0.6 were omitted (run-throughs). The orange dots represent the results related to a constant upper stress level without consideration of the temperature effects (planned $S_{c,max}$). The red dots show the upper stress level taking into account the maximum temperature $S_{c,max}(T_{max})$; the blue ones are calculated values for the upper stress level considering the mean temperature $S_{c,max}(T_{mean})$. Linear trend lines were added to all series.

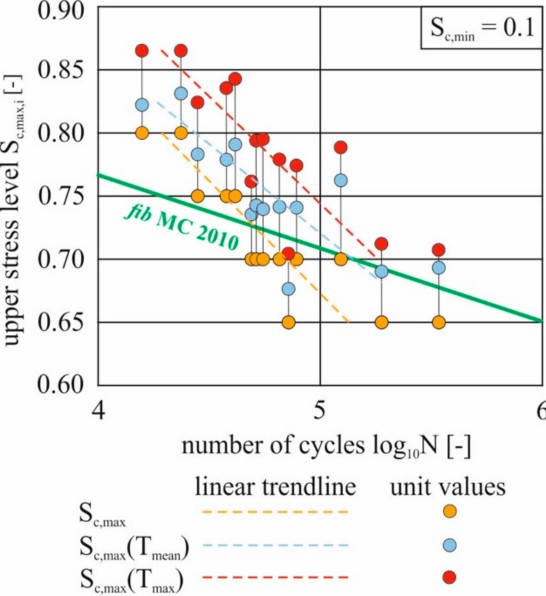

**Figure 6.** Numbers of load cycles N achieved in the experiments and associated linear trend lines, displayed according to the maximum stress level as scheduled and to maximum stress levels adjusted to the maximum measured temperature as well as the average temperature during the tests.

Both approaches led to movement of the measured values vertically upwards. The values with regard to $S_{c,max}(T_{max})$ represented almost a parallel shift of the trend line without adjustment (reference stress level $S_{c,max}$). The red trend line lies above the Wöhler

curve according to fib Model Code 2010 [10], but its course is much steeper. This means that the maximum measured temperature approach in the experiment only showed a good approximation of the lower maximum stresses (0.70 and 0.65); as the maximum stress increased, this tended to lead to an overestimation of the test results. This seemed logical when considering the strength curves in Figure 4. With lower maximum stress levels, a regressive course can be seen, which leads accordingly to a degressive course of the referred maximum stress. The test specimen therefore only experienced the planned initial stress range for a very short time over the duration of the attempt and then a related stress close to the value of the upper stress at maximum temperature for a significantly longer time. With high maximum stresses as initial values, on the other hand, a continuous steep gradient can be seen. In the case of an almost linear decrease in concrete strength with increasing concrete temperature, the approach of the lowest strength $f_c(T_{max})$ achieved was too strong a reduction.

The reduction by using the mean value $S_{c,max}(T_{mean})$ determined over the entire test seemed to be more meaningful. The generated trend line lies in the high upper stress range correspondingly closer to the unmodified entry value, and with decreasing influence of temperature near the maximum value the correction influence increases. However, at 0.65 the fib Model Code 2010 line is undercut in 2 of 3 cases and the generated trend line is still too steep. Another disadvantage of this approach was that the determination of the mean temperature increase required continuous data recording and the evaluation of the temperature curves. However, the aim was to work only with the maximum value of the temperature in order to keep the evaluation simple when applying the method in many experiments.

In addition to the temperature and the maximum stress level, the minimum stress level $S_{c,min}$ should also be included, so the method can also be used for minimum stress levels other than those previously discussed. Accordingly, the stress range $\Delta S$ is the most sensible input value. Based on the experimental results, at $\Delta S = 0.55$, a calculated temperature-dependent maximum stress should be calculated in the range of the maximum temperature. This value was chosen as the limit value because, on the one hand, the undercutting of the Wöhler curve is strongest here and, on the other hand, at 0.55 or smaller stress play the degressive course becomes increasingly clear and thus the temporal part of the high temperature increases. As the stress range increases, the influence of the measured maximum temperature must decrease, because the curves approach a more linear increase until they even increase exponentially from about half the test duration onwards in the case of very large stress plays. For this purpose, a reduction factor $\beta_{cor}$ was introduced, which can be calculated with Equation (4).

$$\beta_{cor} = (S_{c,max} - S_{c,min})/0.55 \geq 1.0 \tag{4}$$

For stress ranges smaller than 0.55, the limit 1.0 was defined, as the curve approaches more and more the maximum value of the temperature, but the strength decrease can never be larger than that at the measured maximum temperature $T_{max}$. In addition, below 0.55 a small influence of the heating can generally be expected owing to a small increase in temperature. The correction factor $\beta_{cor}$ is then considered when determining the corrected maximum stress level as in Equation (5):

$$S_{c,max,cor}(T) = S_{c,max} + [(S_{c,max}/(f_c(T_{max})/f_{cm}) - S_{c,max})/(1 + (\beta_{cor} - 1) * 10]. \tag{5}$$

## 3. Results

The presented method was now applied to the database. First, the tests at 10 Hz and a minimum stress level of 0.1 were considered (see Figure 6). Figure 7 shows the measured numbers of load cycles for the planned initial upper stress levels $S_{c,max}$ as well as for the calculated value $S_{c,max,cor}(T)$ according to Equations (4) and (5).

Equations (4) and (5) led to a smaller modification of the upper stress level $S_{c,max,cor}(T)$ with increasing stress range $\Delta S$. Thus, the tests with $S_{c,max} = 0.8$ as the maximum intended

upper initial stress level were only minimally modified. With increasing test durations with lower stress plays and, thus, longer temperature-related strength reduction, a larger correction became visible. The course of the calculated trend line for the corrected values (blue) shows exactly the desired change compared to the course of the trend line of the initial values (yellow) without temperature consideration. Comparing the modified test results with the Wöhler curve according to fib Model Code 2010 [10], one can see that the corrected tested maximum stress level was everywhere so high that the number of load cycles lay above it. Figure 8 shows the verification of the proposed method with changed minimum stress levels $S_{c,min}$ in order to verify whether the stress-range-dependent equation also provided plausible results in these cases.

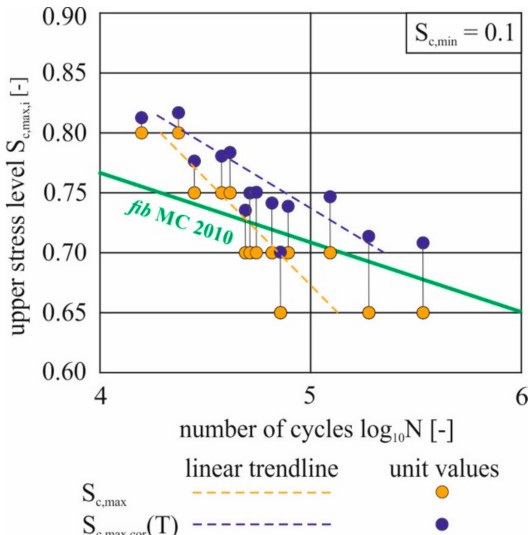

**Figure 7.** Achieved numbers of load cycles in relation to the planned upper stress levels $S_{c,max}$ as well as to the temperature-corrected upper stress levels $S_{c,max,cor}(T)$ according to Equations (4) and (5) with associated linear trend lines; $f$ = 10 Hz and $S_{c,min}$ = 0.1.

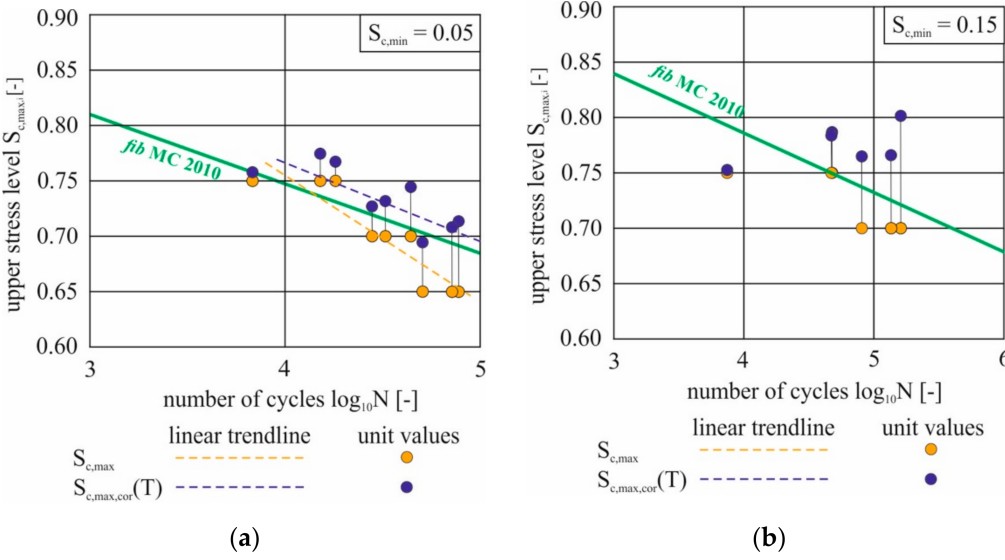

**Figure 8.** Achieved numbers of load cycles in relation to the initial planned upper stress level $S_{c,max}$ as well as the corrected upper stress level $S_{c,max,cor}(T)$ according to Equations (4) and (5) with corresponding linear trend lines; $f$ = 10 Hz, $S_{c,min}$ = 0.05 (**a**) and 0.15 (**b**), respectively.

The values shown in diagram (a) were achieved with $S_{c,min}$ = 0.05, which was a common initial value in other research projects, e.g., in the framework of the Priority

Programme SPP 2020 [23–25]. Here, too, the trend line for the original input values $S_{c,max}$ is steeper than the Wöhler curve. Consideration of the temperature development in the concrete specimen led to a shift and changed inclination of the trend line, which corresponds to the expected picture. For $S_{c,min} = 0.15$ (diagram (b)), the usable database was very small. The samples tested with $S_{c,max} = 0.65$ showed only small heating and did not fail within 2 million load cycles. The test values generated with $S_{c,max} \geq 0.7$ showed the expected behaviour, taking into account the correction value presented. There was only one sample that, contrary to the general trend, failed very early and without significant heating. It can be concluded that the method works for different stress ranges without being bound to one minimum stress level.

The results presented so far were generated at a frequency $f$ of 10 Hz, as the largest database was available here. To verify the method for different frequencies, the other tests from Table 1 at $f = 3 \dots 30$ Hz were also considered. In Figure 9, all test values are shown except for the run-throughs, which occurred mainly in the series with $f = 3$ Hz or $\Delta S < 0.55$.

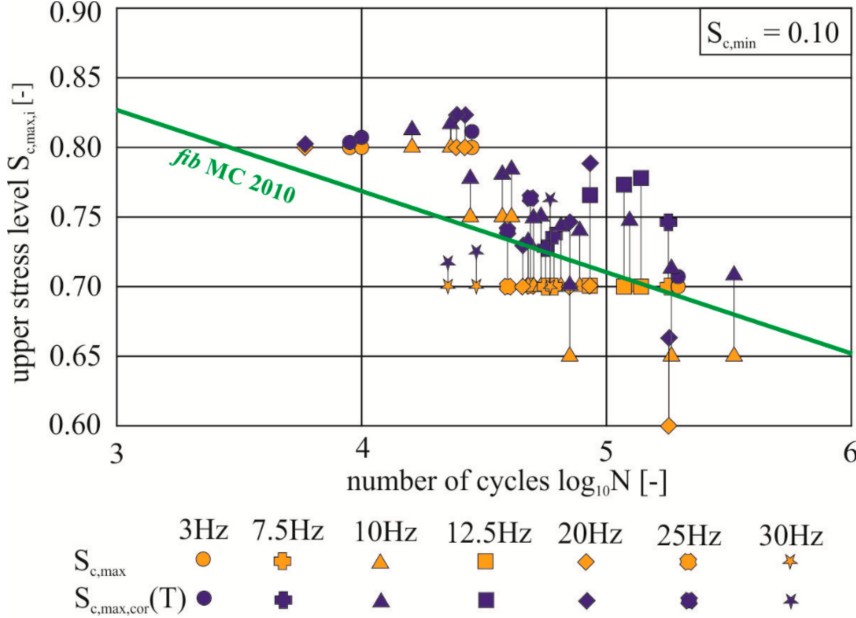

**Figure 9.** Achieved number of load cycles with initial maximum stress as well as the corrected maximum stress level according to Equations (4) and (5) at different frequencies from 3 to 30 Hz and $S_{c,min} = 0.10$.

Because of the large number of trials and parameter combinations, there was a wide range within the test data, which is why separate trend lines are not shown. Even without these, some conclusions can be drawn. For $S_{c,max} = 0.7$, examined at six different frequencies, the number of load cycles tended to decrease with increasing test speed. This was consistent with what has been said so far. When considering the changing material temperature according to the proposed method, most of the test values in Figure 9 are situated above the Wöhler curve according to fib Model Code 2010 [10]. At a top stress level of $S_{c,max} = 0.8$, as expected, no significant temperature-related influence was detected. Accordingly, the adjustments in these tests were very low. At a nominal upper stress $S_{c,max} = 0.6$, all test samples reached 2 million load cycles without failure—with the exception of one test specimen that failed prematurely after loading at 20 Hz, so that a frequency-related evaluation was not possible.

The Wöhler line according to fib Model Code 2010 [10] represents a logarithmic mean. Accordingly, the partial drop below the line was considered unproblematic even for the corrected values if the mean value did not fall significantly below the line. In order to check this, the expected numbers of load cycles according to Model Code 2010 [10] were determined (a) for the initially defined upper stress level $S_{c,max}$, and (b) for the modified

value $S_{c,max,cor}(T)$, and then compared with the achieved numbers of load cycles in the experiments for the mean values for each load configuration, see Table 3.

**Table 3.** Result overview of the presented correction method based on average values of the maximum stresses and the required number of load cycles for all series without run-throughs.

| Frequency | Stress Level | | | Load Cycles | | |
|---|---|---|---|---|---|---|
| $f$ (Hz) | $S_{c,max}$ (−) | $S_{c,max,cor}(T)$ (−) | $S_{c,min}$ (−) | Expected for $S_{c,max}$ (−) | Expected for $S_{c,max,cor}(T)$ (−) | Achieved (−) |
| 3 | 0.80 | 0.805 | 0.10 | 2772 | 2052 | 15,776 |
| 7.5 | 0.70 | 0.722 | 0.10 | 145,938 | 35,828 | 98,965 |
| 10 | 0.65 | 0.705 | 0.05 | 357,145 | 50,708 | 67,048 |
| | | 0.708 | 0.10 | 1,058,919 | 109,642 | 198,391 |
| | 0.70 | 0.735 | 0.05 | 57,489 | 16,939 | 35,199 |
| | | 0.744 | 0.10 | 145,938 | 26,833 | 69,780 |
| | | 0.776 | 0.15 | 380,939 | 17,922 | 124,579 |
| | 0.75 | 0.767 | 0.05 | 9254 | 5230 | 13,415 |
| | | 0.780 | 0.10 | 20,133 | 6040 | 35,484 |
| | | 0.774 | 0.15 | 44,742 | 19,986 | 33,463 |
| | 0.80 | 0.815 | 0.10 | 2772 | 1521 | 19,659 |
| 12.5 | 0.70 | 0.731 | 0.10 | 145,938 | 8819 | 115,620 |
| 20 | 0.70 | 0,723 | 0.10 | 145,938 | 21,704 | 67,665 |
| | 0.80 | 0.807 | 0.10 | 2772 | 1602 | 18,779 |
| 25 | 0.70 | 0.723 | 0.10 | 145,938 | 21,714 | 44,626 |
| 30 | 0.70 | 0.734 | 0.10 | 145,938 | 48,222 | 36,782 |

The tabular compilation of the results shows that the achieved load cycles were closer to the expected values according to fib Model Code 2010 [10] when the temperature correction was taken into account than without this modification. After applying the correction method, only the mean value of the 30 Hz series still fell below the expected value. Without modification, this still occurred for 11 of 16 configurations. The sometimes very significant exceedances after correction of the top stress in individual configurations did not represent an overestimation of the concrete, if one considers that the tests without significant heating (with $f$ = 3 Hz) also lay above the expected value for $S_{c,max}$ = 0.7 with 2 million load cycles without failure. A further safety margin was given by using the maximum temperature at the concrete surface for modification. As shown in [12,15], the sample's core temperature was even higher because of the permanent heat dissipation via the surface. Consequently, the strength reduction inside the specimen is sometimes even higher than considered in this calculation.

## 4. Summary, Conclusions and Outlook

The findings presented in this paper were the result of the research project "Influence of load-induced temperature fields on the fatigue behaviour of UHPC subjected to high frequency compression loading", which was part of the DFG Priority Programme "SPP 2020—Cyclic deterioration of High-Performance Concrete in an experimental-virtual lab" [23].

At the beginning, our own cyclic tests on a UHPC were presented, including load configuration, achieved number of load cycles, and measured maximum surface temperature. In [13,16,21], the main parameters influencing the material heating were already discussed. Based on the investigation of the influence of specimen temperature on the static compressive strength [22], the change in the applied load level over the course of a cyclic trial was derived; an influence that cannot be neglected and that reaches into the two-digit percentage range. The thesis was that this is the main reason for the sometimes

significant drop below the Wöhler curve according to fib Model Code 2010 [10]. Instead of changing the testing concept, for example, by progressive change in frequency or cooling breaks when predefined limit temperatures were reached, the aim was to develop an evaluation method that considered the relationship between reference compressive strength and material temperature, as the "real" stress ratio changes depending on the material heating and the resulting change in reference strength. If this relationship was known, the test results could be corrected subsequently and entered into the Wöhler curve diagrams.

For the correction, an approximation formula for the analysed UHPC was developed that determines how strongly the maximum measured temperature is considered depending on the stress range. As a basis, the adjustments with the maximum as well as the mean value of the temperature were used. With a degressive course of the temperature development and thus an increasing temporal influence of the maximum strength reduction, the influence of the maximum temperature increased with the presented evaluation method.

The correction of all test results by means of the presented evaluation method led to an adjustment of the trend lines in the Wöhler curve diagram, which now came close to the expected course according to fib Model Code 2010 [10]. This made it possible to correct the dent in the Wöhler curve that occurred in cyclic tests of UHPC at a stress play $\Delta S = 0.50 \ldots 0.65$ and test frequencies $f > 3$ Hz in retrospect.

Using further data sets, the applicability of the presented method was shown for different load plays and test frequencies between 3 and 25 Hz. In the future, it will therefore be possible to efficiently test high-performance concretes at high test speeds and, subsequently, to mathematically extract the associated influence of heating. The only thing required is about a measurement of the temperature-dependent change in the static compressive strength, which is specified in fib Model Code 2010 [10] up to HPC and which was determined separately for the examined UHPC [22]. Based on the maximum temperature measured at half the specimen height, after failure the test result can be entered in the Wöhler curve diagram with the corrected upper stress level $S_{c,max,cor}(T)$. Thus, several values are no longer obtained at a constant upper stress level $S_{c,max}$, but this is exactly what happens owing to the change in concrete strength during the pressure fatigue tests. However, if many attempts are entered in the logarithmic diagrams, the result is just as large a database, which makes it possible to create a Wöhler curve.

The method was tested for many different load configurations, but a verification for additional concretes, e.g., regarding differing concrete compositions, strengths, etc., still must be carried out to prove the general validity of the formulated equations. Deviating specimen geometries should also be considered. The diameter influences the ratio between core and surface temperature, whereby no linear relationship between these parameters is expected. In principle, however, an approach is presented in this publication that will enable efficient testing of high-performance concretes in the future without causing an unwanted underestimation of the material at the fatigue strength.

**Funding:** This research was funded by the German Research foundation (DFG). The investigations were carried out in the project 'Influence of load-induced temperature fields on the fatigue behaviour of UHPC subjected to high frequency compression loading' (funding period: September 2017–December 2021), project number: 353981739 (experimental part: SCHE 1966/1-1) as part of the DFG Priority Programme 'SPP 2020—Cyclic deterioration of High-Performance Concrete in an experimental-virtual lab' [23].

**Institutional Review Board Statement:** Not applicable.

**Informed Consent Statement:** Not applicable.

**Acknowledgments:** First, I would like to thank my project managers Silke Scheerer and Ngoc Linh Tran, who as project initiators made it possible for me to work on this topic. Furthermore, my thanks go to my colleagues in the Otto Mohr Laboratory in the Institute of Concrete Structures, TU Dresden, where the specimens were produced and the main part of the experiments took place. The presented correction method is based also on tests carried out in cooperation with the Materials Testing Institute of the University of Stuttgart and the Institute for Modelling and Simulation—Construction of the

Bauhaus University Weimar, to whom I would like to offer my thanks as well. Additional thanks for their cooperation go to the colleagues and partners in the SPP research projects and the coordination project of the SPP 2020.

**Conflicts of Interest:** The author declares no conflict of interest.

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
