# Peer review of "Consideration of the Heating of High-Performance Concretes during Cyclic Tests in the Evaluation of Results"

_2673-3161, doi:10.3390/applmech2040044_

Round 1

Reviewer 1 Report

Reviewer’s Comments to the Author

General

The author presents an investigation of a material made of high-performance compressive concrete.

The paper needs to improve to have an acceptable technical level mainly in terms of the definition of concrete compressive strength based on the design codes (i.e., ACI) and the concrete categories, then high-performance concrete specification in Introduction. In addition, the English writing level of the paper and the technical explanations should be improved considering the below suggestions.

Technical and Editorial

  1. The paper brought up an investigation related to the temperature effects that can be subsequently corrected during the analysis of the results. The author should add some sentences with updated references related to the heating and high-performance concrete specifications.

  1.   In title and line 20: Which references were used for finding 200 MPa as a strength.

  1.  Line 71: The sentence “Only the tests with more than 20 Hz were carried out in Weimar at the testing laboratory of the Institute of Structural Engineering.” It is not clear, Weimar, Germany?

  1. In section 4 (Summary, conclusion, and outlook), the discussion and the results are reprinted with some references. It seems that the main conclusion is not clear enough. The outer can present another part with enough clarification as a conclusion or explain the idea that supports the 4th section.

Reviewer 2 Report

Please find the attached pdf file.

Reviewer 3 Report

I find this paper valuable for the field and worth of acceptance. Please review my short comments regarding your manuscript:

Comment 1: It would be better to take the equation from Figure 2 out and introduce it as one of the numbered equations.

Comment 2: The value of 0.55 for deltaS seems somewhat arbitrary, the paper would benefit of evidence for this value.

Comment 3: The temperature inside the concrete does not increase linearly with specimen size, thus, this method should be clearly stated valid only for this specific geometry of samples.

Reviewer 4 Report

The manuscript entitled "Consideration of the heating of high-performance concretes during cyclic tests in the evaluation of results” showed how the temperature affects load cycles achieved in high-speed tests compared to tests carried out at slow speeds by taking into account the applied stress as well as the maximum temperature reached.

The author presented an analysis of existing experimental results. The author should address the following technical comments before any further process.

Technical comments:

  • The English writing of the manuscript needs improvement. Therefore, it could benefit greatly from professional editing to improve technical writing and English.
  • The abstract is very poor. The used methodology and the main conclusion should be highlighted in the abstract. So, the reader can understand what will be in the manuscript. Also, there are a few typos in the abstract that should be corrected.
  • The authors should increase their discussion on previous related research and highlight how their study is providing a different approach or adding significantly to what has been done.
  • Lines 12-13: Unclear sentence!
  • Figure 1: Is it a concrete specimen in Fig. 1b? It looks like a steel specimen. Also, there is no need to provide these kinds of figures because the author did not provide an experimental investigation. All experimental results were presented from other references.
  • Line 92: It should be "In all these cases".
  • Line 113: Figure 3 shows the range was starting with 20, not -20. The authors should clarify this.
  • Line 126: Figure 3 is talking about the static compressive strength of concrete. However, the author in this section is talking about cyclic tests. So, is Figure 3 is correct to be mentioned in this section?
  • Lines 129-130: How is the equation in Fig. 3 used during the test (for cyclic tests)? It should be for static tests without cyclic loading.
  • Line 137: What is the unit of the number "200,000". The author said "load changes up to 200,000". So, it should be 200,000 N or kN. Or what? Or do you mean the number of cycles? The author should clarify this through the manuscript.
  • Lines 331-333: Unclear sentence!
  • Line 353: Is it 30 Hz as mention in line 275 or 25 Hz as mentioned here?
  • This reviewer believes that the only contribution from the author in this manuscript is section 2.6. There is no significant contribution. The author should validate the proposed equation with the experiments and shows how much this matches. I mean statistical analysis between the experimental results and the developed results.
  • The author should expand the practical applications of the proposed equations in civil engineering. In real concrete structures, concrete is under the effect of low frequency like bridges or even concrete buildings under the effect of an earthquake. I suggest adding a section to discuss the effect of induced temperature from the cyclic tests on the compressive strength of concrete.

Round 2

Reviewer 4 Report

The authors addressed most of the reviewer's comments and it can be accepted for publication.